# Metabolic biomarker profiling for identification of susceptibility to severe pneumonia and COVID-19 in the general population

Heli Julkunen[1], Anna Cichońska[1], P Eline Slagboom[2,3], Peter Würtz[1]*, Nightingale Health UK Biobank Initiative[1]

[1]Nightingale Health Plc, Helsinki, Finland; [2]Molecular Epidemiology, Department of Biomedical Data Sciences, Leiden University Medical Center, Leiden, Netherlands; [3]Max Planck Institute for Biology of Ageing, Cologne, Germany

**Abstract** Biomarkers of low-grade inflammation have been associated with susceptibility to a severe infectious disease course, even when measured prior to disease onset. We investigated whether metabolic biomarkers measured by nuclear magnetic resonance (NMR) spectroscopy could be associated with susceptibility to severe pneumonia (2507 hospitalised or fatal cases) and severe COVID-19 (652 hospitalised cases) in 105,146 generally healthy individuals from UK Biobank, with blood samples collected 2007–2010. The overall signature of metabolic biomarker associations was similar for the risk of severe pneumonia and severe COVID-19. A multi-biomarker score, comprised of 25 proteins, fatty acids, amino acids, and lipids, was associated equally strongly with enhanced susceptibility to severe COVID-19 (odds ratio 2.9 [95%CI 2.1–3.8] for highest vs lowest quintile) and severe pneumonia events occurring 7–11 years after blood sampling (2.6 [1.7–3.9]). However, the risk for severe pneumonia occurring during the first 2 years after blood sampling for people with elevated levels of the multi-biomarker score was over four times higher than for long-term risk (8.0 [4.1–15.6]). If these hypothesis generating findings on increased susceptibility to severe pneumonia during the first few years after blood sampling extend to severe COVID-19, metabolic biomarker profiling could potentially complement existing tools for identifying individuals at high risk. These results provide novel molecular understanding on how metabolic biomarkers reflect the susceptibility to severe COVID-19 and other infections in the general population.

*For correspondence:
peter.wurtz@nightingalehealth.com

## Introduction

The coronavirus disease 2019 (COVID-19) pandemic affects societies and healthcare systems worldwide. Protection of those individuals who are most susceptible to a severe and potentially fatal COVID-19 disease course is a prime component of national policies, with stricter social distancing and other preventative means recommended mainly for elderly people and individuals with pre-existing disease conditions. The prominent susceptibility to severe COVID-19 for people at high age has been linked with impaired immune response due to chronic inflammation caused by ageing processes (*Akbar and Gilroy, 2020*). However, large numbers of seemingly healthy middle-aged individuals also suffer from severe COVID-19 (*Zhou et al., 2020*; *Atkins et al., 2020*; *Williamson et al., 2020*); this could partly be due to similar molecular processes related to impaired immunity. A better understanding of the molecular factors predisposing to severe COVID-19 outcomes may help to explain the risk elevation ascribed to pre-existing disease conditions. From a translational point of view, this might also complement the identification of highly susceptible individuals in general population settings beyond current risk factor assessment.

**eLife digest** National policies for mitigating the COVID-19 pandemic include stricter measures for people considered to be at high risk of severe and potentially fatal cases of the disease. Although older age and pre-existing health conditions are strong risk factors, it is poorly understood why susceptibility varies so widely in the population.

People with cardiometabolic diseases, such as diabetes and liver diseases, or chronic inflammation are at higher risk of severe COVID-19 and other infections including pneumonia. These conditions alter the molecules circulating in the blood, providing potential 'biomarkers' to determine whether a person is more likely to develop a fatal infection. Uncovering these blood biomarkers could help to identify people who are prone to life-threatening infections despite not having ever been diagnosed with a cardiometabolic disease.

To find these biomarkers, Julkunen et al. studied blood samples that had been collected from 105,000 healthy individuals in the United Kingdom over ten years ago. The data showed that individuals with biomarkers linked to low-grade inflammation and cardiometabolic disease were more likely to have died or been hospitalised with pneumonia.

A score based on 25 of these biomarkers provided the best predictor of severe pneumonia. This biomarker score performed up to four times better within the first few years after blood sampling compared to predicting cases of pneumonia a decade later. The same blood biomarker changes were also linked with developing severe COVID-19 over ten years after the blood samples had been collected. The predictive value of the biomarker score was similar for both severe COVID-19 and the long-term risk of severe pneumonia.

Julkunen et al. propose that the metabolic biomarkers reflect inhibited immunity that impairs response to infections. The results from over 100,000 individuals suggest that these blood biomarkers may help to identify people at high risk of severe COVID-19 or other infectious diseases.

Pneumonia is a life-threatening complication of COVID-19 and the most common diagnosis in severe COVID-19 patients. As for COVID-19, the main factors that increase the susceptibility for severe community-acquired pneumonia are high age and pre-existing respiratory and cardiometabolic diseases, which can weaken the lungs and the immune system (*Almirall et al., 2017*). Based on analyses of large blood sample collections of healthy individuals, biomarkers associated with the risk for severe COVID-19 are largely shared with the biomarkers associated with the risk for severe pneumonia, including elevated markers of impaired kidney function and inflammation and lower HDL cholesterol (*Ho et al., 2020*). This may indicate that these molecular markers may reflect an overall susceptibility to severe complications after contracting an infectious disease.

Comprehensive profiling of metabolic biomarkers, also known as metabolomics, in prospective population studies have suggested a range of blood biomarkers for cardiovascular disease and diabetes to also be reflective of the susceptibility for severe infectious diseases (*Ritchie et al., 2015*; *Deelen et al., 2019*). Metabolic profiling could therefore potentially identify biomarkers that reflect the susceptibility to severe COVID-19 among initially healthy individuals. However, such studies require measurement of vast numbers of blood samples collected prior to the COVID-19 pre-pandemic. Conveniently, a broad panel of metabolic biomarkers have recently been measured using nuclear magnetic resonance (NMR) spectroscopy in over 100,000 plasma samples from the UK Biobank.

Here, we examined if NMR-based metabolic biomarkers from blood samples collected a decade before the COVID-19 pandemic associate with the risk of severe infectious disease in UK general population settings. Exploiting the shared risk factor relation between susceptibility to severe COVID-19 and pneumonia (*Ho et al., 2020*), we used well-powered statistical analyses of biomarkers with severe pneumonia events to develop a multi-biomarker score that condenses the information from the metabolic measures into a single multi-biomarker score. Taking advantage of the time-resolved information on the occurrence of severe pneumonia events in the UK Biobank, we mimicked the influence of the decade lag from blood sampling to the COVID-19 pandemic on the biomarker associations, and used analyses with short-term follow-up to interpolate to a scenario of identifying individuals susceptible to severe COVID-19 in a preventative screening setting. Our

primary aim was to improve the molecular understanding on how metabolic risk markers may contribute to increased predisposition to severe COVID-19 and other infections.

## Results

A flow diagram of eligible study participants and case numbers is shown in *Figure 1*. Clinical characteristics of the study population are listed in *Table 1*. Among the 105,146 UK Biobank study participants with complete data on metabolic biomarkers and severe pneumonia outcomes, and no prior history of diagnosed pneumonia, there were 2507 severe pneumonia events recorded in hospital or death registries after the baseline blood sampling (median follow-up time 8.1 years).

For the severe COVID-19 analyses, there were 652 PCR-confirmed positive cases diagnosed in hospital (inferred as severe cases in this study) among the 92,725 individuals with COVID-19 data linkage available per 3rd of February 2021. The number of severe COVID-19 cases in the UK Biobank closely followed the trends in hospitalised individuals for COVID-19 in England (*Figure 1—figure supplement 1*). In February 2021, the age range of study participants was 49–84 years. The median duration from blood sampling to the COVID-19 pandemic was 11.2 years (interquartile range 10.0–12.6). The prevalence of chronic respiratory and cardiometabolic diseases was similar for study

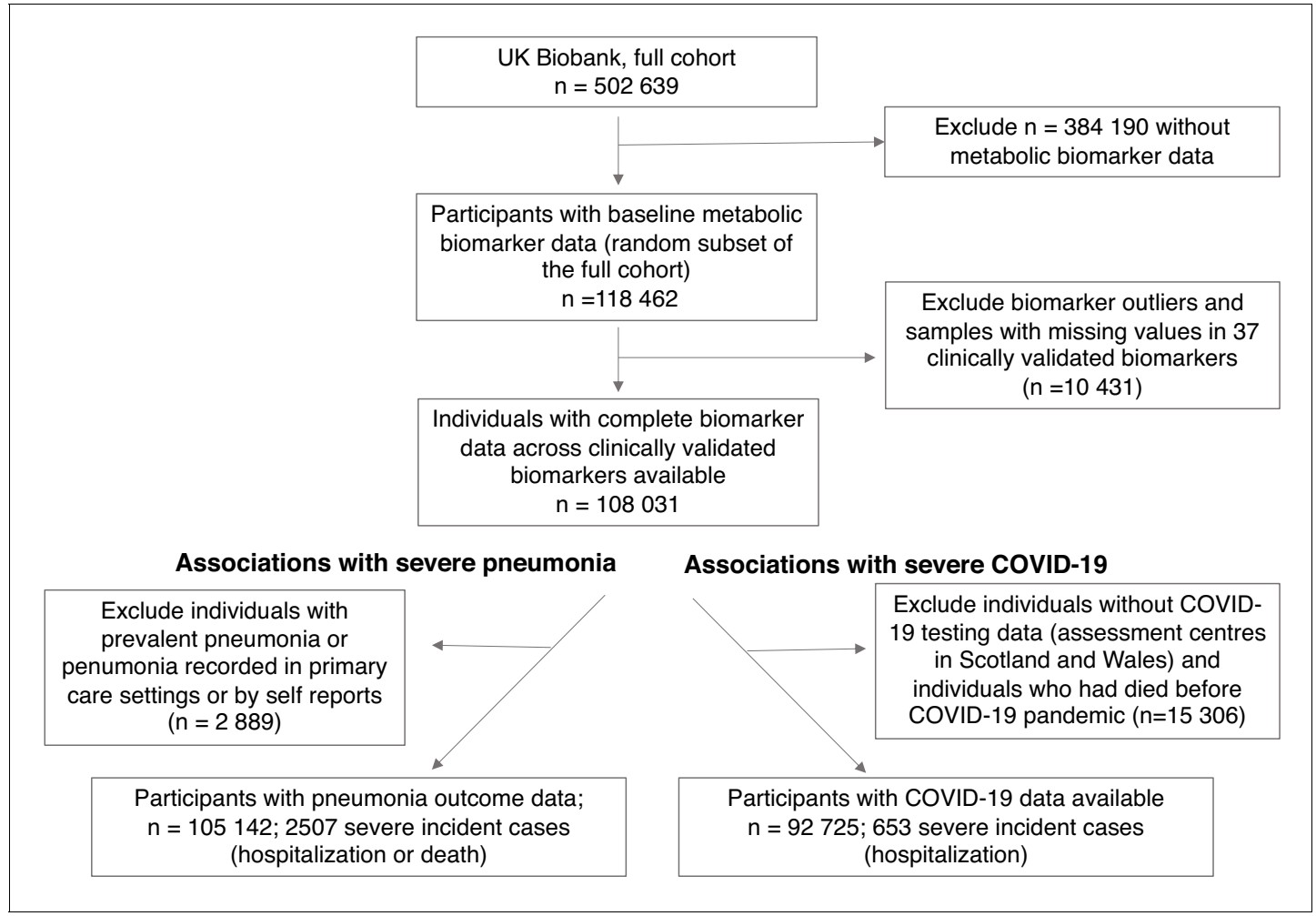

**Figure 1.** Flow diagram of study participants and case numbers. Overview of eligible study participants for the analysis of metabolic biomarkers for the susceptibility to severe pneumonia and COVID-19 in the UK Biobank. Case and control definitions are described in Materials and methods.

The online version of this article includes the following figure supplement(s) for figure 1:

**Figure supplement 1.** Numbers of COVID-19 positive and hospitalised individuals in the UK Biobank and the whole of England during the course of the COVID-19 pandemic.

**Table 1.** Clinical characteristics of the UK Biobank participants in the current study.

| | Severe pneumonia (diagnosis in hospital or death record) | | Severe COVID-19 (diagnosis in hospital) | |
| --- | --- | --- | --- | --- |
| | Incident cases | Controls | Incident cases | Controls |
| Individuals with NMR biomarker measures | 2507 | 102 639 | 652 | 92 073 |
| Age at blood sampling (median, [range]) | 62 [40-70] | 58 [39-70] | 60 [40-70] | 58 [39-70] |
| Females (%) | 44% | 54% | 43% | 54% |
| Body mass index (mean, kg/m$^2$) | 28.5 | 27.4 | 28.7 | 27.3 |
| **Proportion with prevalent diseases** | | | | |
| Cardiovascular disease (%) | 17.5% | 6.6% | 14.7% | 6.4% |
| Diabetes (%) | 9.3% | 3.9% | 9.2% | 3.8% |
| Lung cancer (%) | 0.4% | 0.1% | 0.3% | 0.1% |
| Chronic obstructive pulmonary disease (%) | 6.1% | 0.7% | 1.8% | 0.8% |
| Liver diseases (%) | 1.5% | 0.7% | 1.7% | 0.7% |
| Renal failure (%) | 3.6% | 1.3% | 2.9% | 1.4% |
| Dementia (%) | 0.1% | 0.01% | 0.0% | 0.01% |

The number of individuals analysed for severe COVID-19 is slightly lower than for severe pneumonia, since COVID-19 data were not available from assessment centres in Scotland and Wales.

participants who developed severe pneumonia and those who contracted COVID-19 and required hospitalisation, with the exception of COPD. There were 33 overlapping cases between severe pneumonia and COVID-19.

## Metabolic biomarkers and severe pneumonia risk

*Figure 2A* shows the associations of 37 biomarkers with severe pneumonia events occurring during the follow-up in the entire study population (n = 105 146). The biomarkers highlighted here are those with a regulatory approval for diagnostics use in the Nightingale Health NMR platform. These biomarkers span most of the different metabolic pathways captured with the NMR platform; results for all 249 metabolic measures quantified are shown in *Figure 2—figure supplements 1–3*. Strong associations were observed across several metabolic pathways: increased plasma concentrations of cholesterol measures, omega-3 and omega-6 fatty acid levels, histidine, branched-chain amino acids and albumin were associated with lower susceptibility to contracting severe pneumonia. Increased concentrations of monounsaturated and saturated fatty acids, as well glycoprotein acetyls (GlycA, a marker of low-grade inflammation) were associated with elevated susceptibility to contracting severe pneumonia.

Since all the biomarkers are quantified in the same single measurement, we examined if even stronger associations with severe pneumonia could be obtained using a combination of multiple biomarkers. We derived this multi-biomarker combination, denoted 'infectious disease score', using logistic regression with LASSO for variable selection, considering the 37 clinically validated biomarkers in a half of the study population as the training set. This resulted in an infectious disease score comprised of the weighted sum of 25 biomarkers, with the weights selected by the machine learning algorithm (*Supplementary file 1*). Broadly similar results were obtained using all 249 metabolic measures quantified in the Nightingale Health NMR platform to derive the multi-biomarker score.

The multi-biomarker infectious disease score was then tested for association with severe pneumonia in the other half of the study population. The magnitude of association for the infectious disease score was approximately twice as strong with severe pneumonia compared to any of the individual biomarkers (*Figure 2B*). The odds for contracting severe pneumonia was increased 67% per 1-SD increment in the infectious disease score. This corresponds to close to fourfold higher risk for

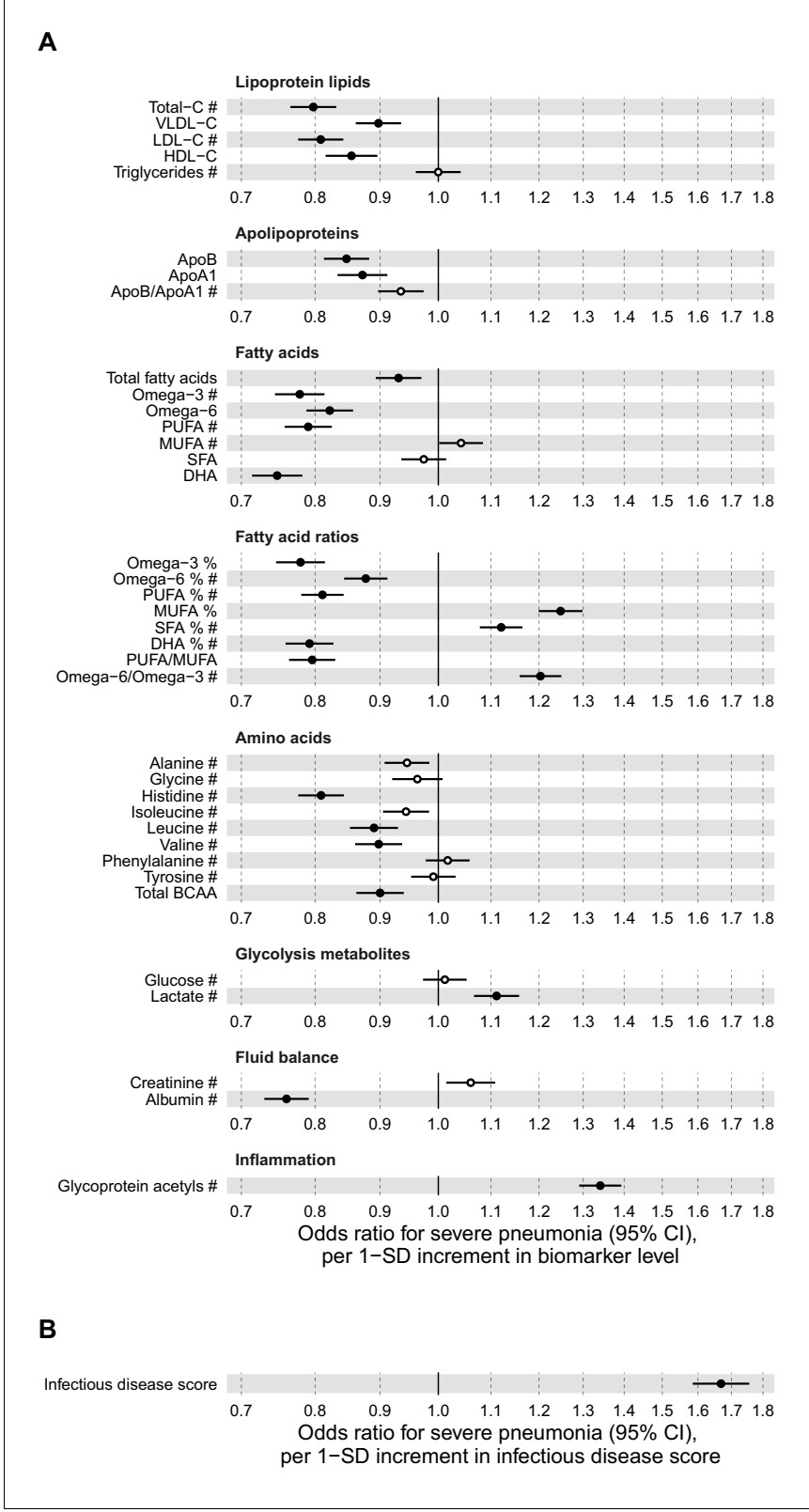

**Figure 2.** Relation of baseline biomarker concentrations to future risk of severe pneumonia in the UK Biobank (n = 105 146; 2507 incident events). (**A**) Odds ratios with severe pneumonia (2507 hospitalisations or deaths during a median of 8 years of follow-up) for 37 clinically validated biomarkers measured simultaneously in a single assay by Nightingale Health NMR platform. (**B**) Odds ratio with severe pneumonia for the multi-biomarker infectious disease score. The infectious disease score comprises of the weighted sum of 25 out of 37 clinically validated biomarkers, optimised for

*Figure 2 continued on next page*

*Figure 2 continued*

association with severe pneumonia based on one half of the study population using LASSO regression. Biomarkers included in the infectious disease score are marked by #. The odds ratio for infectious disease score is evaluated in the other half of the study population (n = 52 573; 1250 events). All models are adjusted for age, sex, and assessment centre. Odds ratios are per 1-SD increment in the biomarker levels. Horizontal bars denote 95% confidence intervals. Closed circles denote p-value<0.001 and open circles p-value≥0.001. BCAA indicates branched-chain amino acids; DHA: docosahexaenoic acid; MUFA: monounsaturated fatty acids; PUFA: polyunsaturated fatty acids; SFA: saturated fatty acids.

The online version of this article includes the following source data and figure supplement(s) for figure 2:

**Source data 1.** Numerical tabulation of odds ratios, betas, standard errors, and p-values for results shown in *Figure 2*.

**Figure supplement 1.** Relation of all biomarkers measured by the Nightingale Health NMR platform to risk of severe pneumonia in UK Biobank (n = 105 146; 2507 events).

**Figure supplement 2.** Relation of all biomarkers measured by the Nightingale Health NMR platform to risk of severe pneumonia in UK Biobank (n = 105 146; 2507 events).

**Figure supplement 3.** Relation of all biomarkers measured by the Nightingale Health NMR platform to risk of severe pneumonia in UK Biobank (n = 105 146; 2507 events).

contracting severe pneumonia among people in the highest quintile of the infectious disease score, compared to those with a score in the lowest quintile.

To assess the robustness of the multi-biomarker score association with severe pneumonia, we adjusted the analyses for prevalent diseases and performed analyses stratified by age and sex (*Figure 3*). The association was attenuated by ~10% in magnitude when adjusting for, or omitting, individuals with a diagnosis of prevalent diseases at time of blood sampling (cardiovascular diseases, diabetes, lung cancer, COPD, liver diseases, renal failure, and dementia; panels 3A and 3B). The association was similar across age groups, and also for men and women analysed separately (panels 3C and 3D).

To mimic the influence of the decade-long lag from blood sample collection to the COVID-19 pandemic, we tested the association of the multi-biomarker infectious disease score with severe pneumonia events occurring during 7–11 years after the blood sampling (*Figure 4A*). Since there were only few severe pneumonia events recorded with more than 9 years of follow-up, we could not fully mimic the decade long time lag to the COVID-19 pandemic. The risk elevation observed in this time-lag accounting scenario was only approximately half of that observed for severe pneumonia events occurring within the first 7 years (odds ratio 1.43 vs 1.75 per 1-SD, respectively; and 2.59 vs 4.27 for individuals in the highest vs lowest quintile of the infectious disease score).

To interpolate to a screening scenario conducted today, we also tested the association with short-term risk of severe pneumonia by analysing events occurring within the first 2 years after the blood sampling (*Figure 4B*). The association magnitude in this analysis of short-term risk scenario was approximately twice as strong as for severe pneumonia events occurring more than 2 years after blood sampling (odds ratio 2.21 vs 1.59 per 1-SD; 7.95 vs 3.35 for individuals in the highest vs lowest quintile of the infectious disease score). The elevated susceptibility to severe pneumonia associated with the multi-biomarker score was therefore three to four times stronger when examining short-term risk as compared to risk of severe pneumonia events occurring almost a decade after the blood sampling.

The elevation in the short-term risk for severe pneumonia for high levels of the infectious disease multi-biomarker score remained strong when adjusting for BMI, smoking and prevalent diseases (odds ratio 6.10 for individuals in the highest vs lowest quintile; *Figure 4—figure supplement 1*).

We further explored the risk gradient for a future onset of severe pneumonia along increasing levels of the infectious disease score, since non-linear effects could potentially facilitate the identification of thresholds for individuals at high susceptibility. *Figure 5A* shows the increase in the proportion of individuals who contracted severe pneumonia according to percentiles of the score. The risk increased prominently in the highest quintile, and particularly for the highest few percentiles. The time-resolved plot of the cumulative probability of severe pneumonia during follow-up is shown in *Figure 5B*. The susceptibility to severe pneumonia was particularly elevated among individuals with the very highest levels of the multi-biomarker infectious disease score. This was observed already during the first few years of follow-up, corroborating the results for long-term and short-term risk shown in *Figure 3*. The prominent and immediate elevation in susceptibility to severe

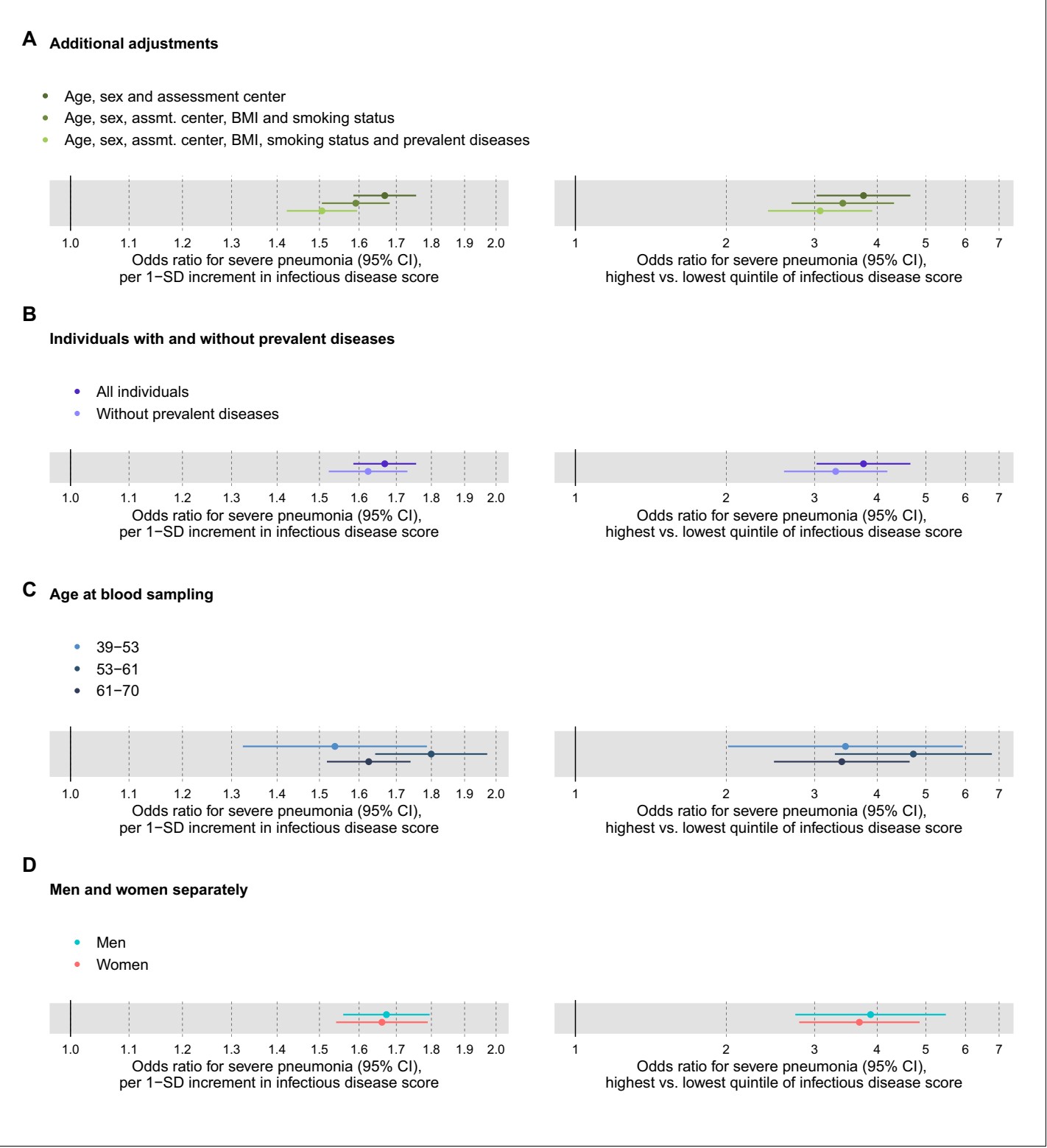

**Figure 3.** Relation of the multi-biomarker infectious disease score to future risk of severe pneumonia with additional adjustments and in subgroups (n = 52 573; 1250 incident events). (**A**) Odds ratios with severe pneumonia after additional adjustments for BMI, smoking status, and prevalent diseases. (**B**) Odds ratios with severe pneumonia in study participants with and without prevalent diseases. (**C**) Odds ratios by age tertiles at the time of blood sampling. (**D**) Odds ratios for men and women separately. All models are adjusted for age, sex, and assessment centre. The left-hand side shows odds ratios per 1-SD increment in the multi-biomarker infectious disease score, and the right-hand side odds ratios for comparing individuals in the highest

*Figure 3 continued on next page*

*Figure 3 continued*

and lowest quintiles of the score. The results are based on the validation half of the study population not used in deriving the infectious disease score (1250 events during a median of 8 years of follow-up).

The online version of this article includes the following source data for figure 3:

**Source data 1.** Numerical tabulation of odds ratios, betas, standard errors, and p-values for results shown in *Figure 3*.

pneumonia was also observed when limiting analyses to individuals without chronic respiratory and cardiometabolic diseases at the time of blood sampling (*Figure 5—figure supplement 1*).

## Metabolic biomarkers and severe COVID-19

*Figure 6* shows the associations of the 37 clinically validated biomarkers and the infectious disease score with the future onset of severe COVID-19 (defined as PCR-confirmed positive inpatient diagnosis). Many of the individual biomarkers had significant associations (p-value<0.001) with increased risk for severe COVID-19. These biomarkers for susceptibility to severe COVID-19 include lower levels of omega-3 omega-6 fatty acids as well as albumin, and higher levels of GlycA. We observed a high concordance in the overall pattern of COVID-19 biomarker associations with severe pneumonia (*Figure 2A*), with a Spearman correlation of 0.89 between the overall biomarker association signatures for severe pneumonia and severe COVID-19 (*Figure 7*).

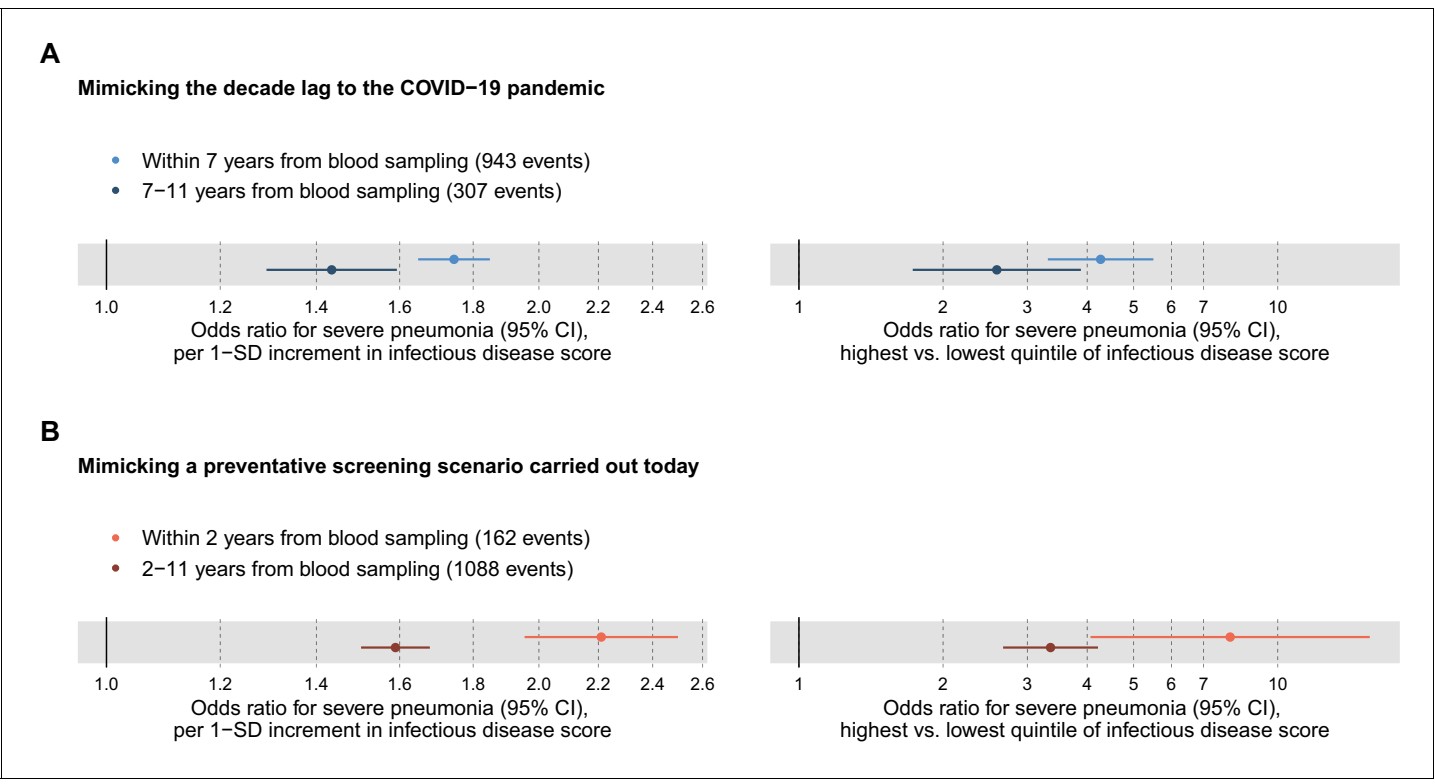

**Figure 4.** Relation of the multi-biomarker infectious disease score to long-term and short-term future risk for severe pneumonia (n = 52 573; 1250 incident events). (**A**) Odds ratios with severe pneumonia events occurring within the first 7 years after the blood sampling, compared to events that occurred 7–11 years after blood sampling. (**B**) Odds ratios for severe pneumonia occurring within and after the first 2 years of blood sampling. Models are adjusted for age, sex, and assessment centre. The left-hand side shows odds ratios per 1-SD increment in the multi-biomarker infectious disease score, and the right-hand side odds ratios for comparing individuals in the highest and lowest quintiles of the score. The results are based on the validation half of the study population that was not used in deriving the infectious disease score.

The online version of this article includes the following source data and figure supplement(s) for figure 4:

**Source data 1.** Numerical tabulation of odds ratios, betas, standard errors, and p-values for results shown in *Figure 4*.

**Figure supplement 1.** Relation of the multi-biomarker infectious disease score to long-term and short-term risk for severe pneumonia after adjustment for risk factors and prevalent diseases (n = 52 573; 1250 events).

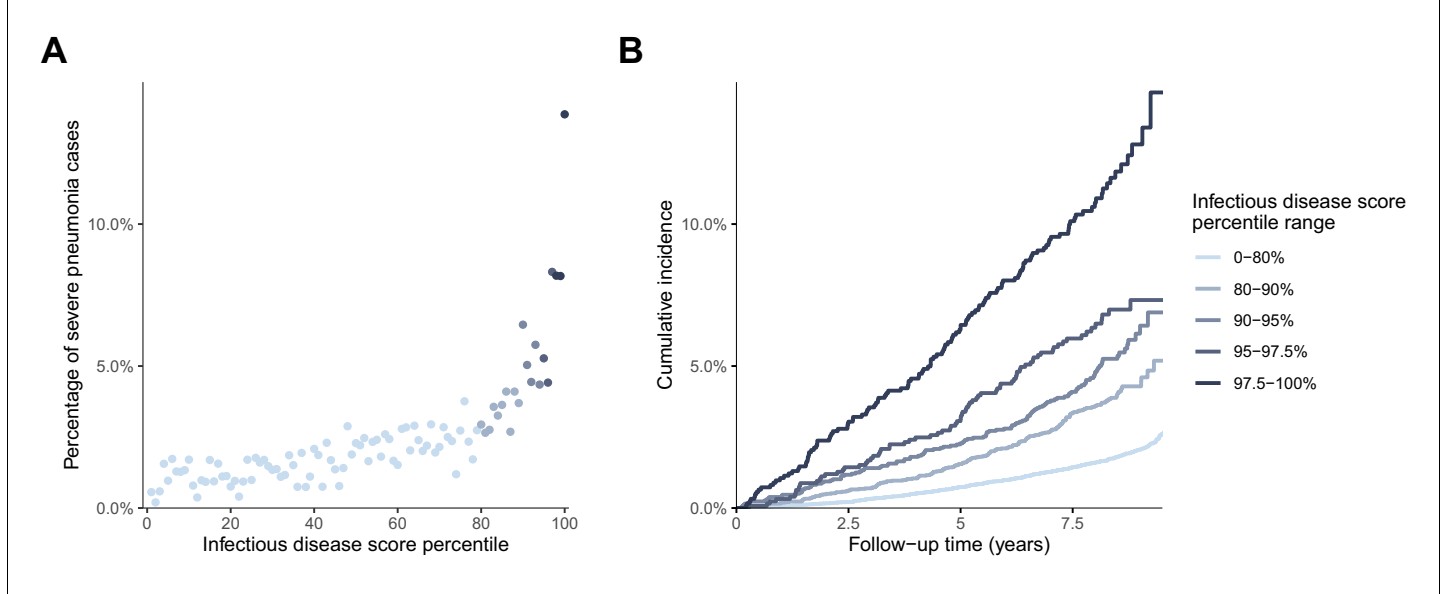

**Figure 5.** Risk gradient for contracting severe pneumonia after the blood sampling according to percentiles of the multi-biomarker infectious disease score (n = 52 573; 1250 incident events). (**A**) Proportion of individuals who contracted severe pneumonia during a median follow-up time of 8.1 years after the blood sampling according to percentiles of the multi-biomarker infectious disease score. Each point represents approximately 500 individuals. (**B**) Kaplan-Meier curves of the cumulative probability for severe pneumonia in quantiles of the multi-biomarker infectious disease score. The follow-up time was truncated at 9.5 years since only a small fraction of individuals were followed longer. Results are based on the validation half of the study population that was not used in deriving the infectious disease score (n = 52,573). The corresponding plots for individuals free of baseline respiratory and cardiometabolic diseases are shown in *Figure 5—figure supplement 1*.

The online version of this article includes the following source data and figure supplement(s) for figure 5:

**Source data 1.** Numerical tabulation of event rates for each percentile in *Figure 5A*.

**Figure supplement 1.** Risk gradients for contracting severe pneumonia by percentiles of the multi-biomarker infectious disease score among individuals without prevalent diseases at time of blood sampling (n = 46,252; 877 events).

The multi-biomarker infectious disease score derived for the future onset of severe pneumonia was also robustly associated with the future onset of severe COVID-19. The odds ratio was 1.40 per 1-SD increment and 2.90 for comparing individuals in the highest quintile of the multi-biomarker infectious disease score to those in the lowest quintile. This magnitude of association with susceptibility to severe COVID-19 was similar to that observed with severe pneumonia events occurring during the interval of 7–11 years after the blood sampling.

We further examined the association of the multi-biomarker infectious disease score with severe COVID-19 after adjustment or exclusion for prevalent diseases, and conducted stratified analyses for age and sex (*Figure 8*). The association with severe COVID-19 was attenuated, but remained significant when adjusted for BMI, smoking and prevalent diseases (panel 7A). The association magnitudes were approximately 20% weaker when limiting the COVID-19 analyses to individuals without prevalent diseases at time of blood sampling (panel 7B). There was no robust evidence of differences in association magnitude according to age (panel 7C) and odd ratios were broadly similar for men and women (panel 7D).

Finally, we examined the technical repeatability and biological stability of measuring the multi-biomarker infectious disease score. The measurement repeatability was high (Pearson correlation 0.94 in blind duplicate samples; *Figure 9A*). Even though the blood samples were primarily non-fasting, the levels of the infectious disease score remained broadly stable during 4 years based on blood samples from repeat visits (Pearson correlation 0.61 between baseline and repeat visit measurements; *Figure 9B*).

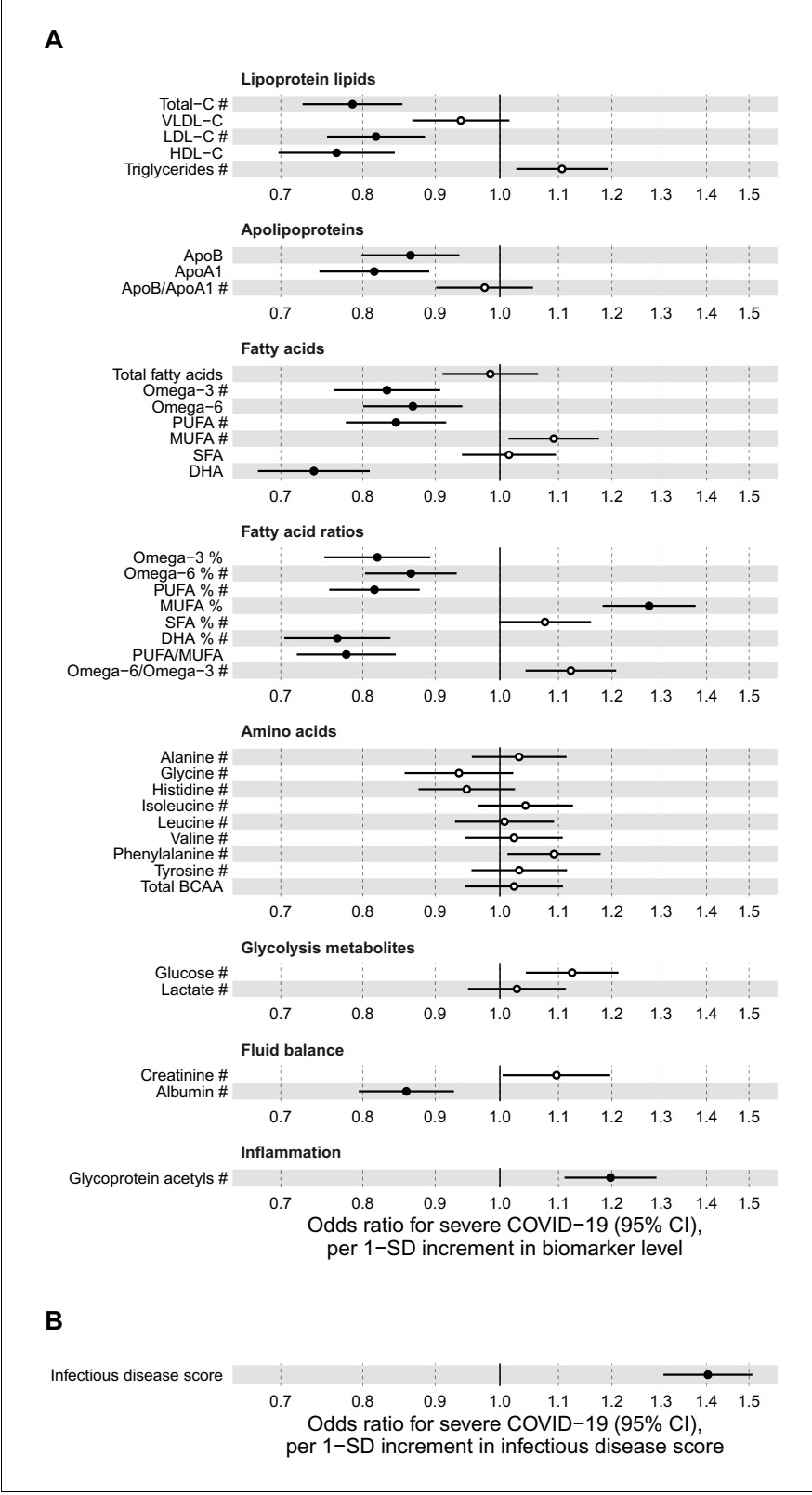

**Figure 6.** Relation of baseline biomarkers and multi-biomarker infectious disease score to future risk of severe COVID-19 (n = 92 725; 652 cases diagnosed in hospital). (**A**) Odds ratios with severe COVID-19 (defined as PCR-positive diagnosis in hospital; 652 cases out of 92 725 individuals) for 37 clinically validated biomarkers measured by NMR. (**B**) Odds ratio with severe COVID-19 for the multi-biomarker infectious disease score. Biomarkers

*Figure 6 continued on next page*

*Figure 6 continued*

included in the infectious disease score are marked by #. Odds ratios are per 1-SD increment in the biomarker levels. Models are adjusted for age, sex, and assessment centre.

The online version of this article includes the following source data for figure 6:

**Source data 1.** Numerical tabulation of odds ratios, betas, standard errors, and p-values for results shown in *Figure 6*.

## Discussion

Most biomarker studies on COVID–19 have focused on characterising already infected patients and their disease prognosis (*Kermali et al., 2020*; *Shen et al., 2020*; *Messner et al., 2020*; *Dierckx et al., 2020*). In contrast, in the largest blood metabolic profiling study to date, we explored biomarker associations for susceptibility to severe pneumonia and COVID-19 in general population settings. We developed a multi-biomarker score for increased susceptibility to a severe infectious disease course, and demonstrated that this biomarker score captures an increased risk for COVID-19 hospitalisation a decade after the blood sampling.

The overall signature of biomarker associations was similar for the susceptibility to severe COVID-19 and to severe pneumonia (*Figure 7*). The proportions of individuals with existing cardiometabolic diseases were also consistent for both of these infectious diseases (*Table 1*). We used these observations of a shared risk factor basis to draw an analogy between susceptibility to severe pneumonia and severe COVID-19, and hereby infer potential implications for preventative screening. We therefore exploited the strong statistical power and time-resolved information on severe pneumonia events for more detailed analyses than was feasible with COVID-19. This led to three important observations. First, the infectious disease multi-biomarker score was largely independent of prevalent chronic respiratory and cardiometabolic diseases (*Figure 3*). Second, the susceptibility to severe pneumonia was drastically elevated in the extreme tail of the multi-biomarker infectious disease score, with 5–10 times higher risk compared to individuals with normal levels of the multi-biomarker score (*Figure 5*). Such features might aid in establishing thresholds for identifying individuals most susceptible to a severe disease course. Third, the odds ratio of the multi-biomarker score for severe pneumonia events occurring after 7–11 years closely matched that of severe COVID-19, for which all events occurred over decade after blood sampling (*Figure 4A*). Yet, screening for the susceptibility to severe COVID-19 would require a strong association with the short-term risk. When confining the analyses of severe pneumonia to events occurring within the first 2 years after blood sampling, the short-term risk elevation was over four times stronger than that observed for long-term risk — individuals with high levels of the multi-biomarker score were almost 7-times more susceptible than people with low levels (*Figure 4B*). If similar enhancement in short-term risk extend to COVID-19, our results could potentially indicate applications for identification of individuals at high susceptibility to a severe COVID-19 disease course. However, the unavailability of metabolic biomarker data from blood samples drawn shortly prior to the pandemic prevents us from examining biomarker associations with short-term COVID-19 susceptibility, and our results should therefore be considered of hypothesis generating nature.

We observed multiple blood biomarkers commonly linked with the risk for cardiovascular disease and diabetes (*Soininen et al., 2015*; *Würtz et al., 2017*; *Holmes et al., 2018*; *Ahola-Olli et al., 2019*) to also be associated with increased susceptibility to both severe pneumonia and severe COVID-19. The biomarkers span multiple metabolic pathways, including low concentrations of lipoprotein lipids, impaired fatty acid balance, decreased amino acid levels and high chronic inflammation. This is the first study to show that many of these blood biomarkers associate with susceptibility to severe infections, potentially indicating that fatty acids and amino acids should not be considered only as biomarkers for cardiometabolic risk. The associations of omega-3 and other fatty acids with the risk for severe COVID-19 may be particularly important, as these measures are more directly modifiable by lifestyle means than common markers of inflammation. The overall pattern of biomarker associations followed a characteristic metabolic signature reflective of an increased susceptibility to a severe infectious disease. This pattern of biomarker associations is broadly similar to what has previously been reported with the risk for all-cause mortality in smaller prospective cohort studies (*Deelen et al., 2019*). It is therefore unlikely that the identified biomarker signature is specific to the risk for severe pneumonia and COVID-19, or even specific to infectious diseases in general. We

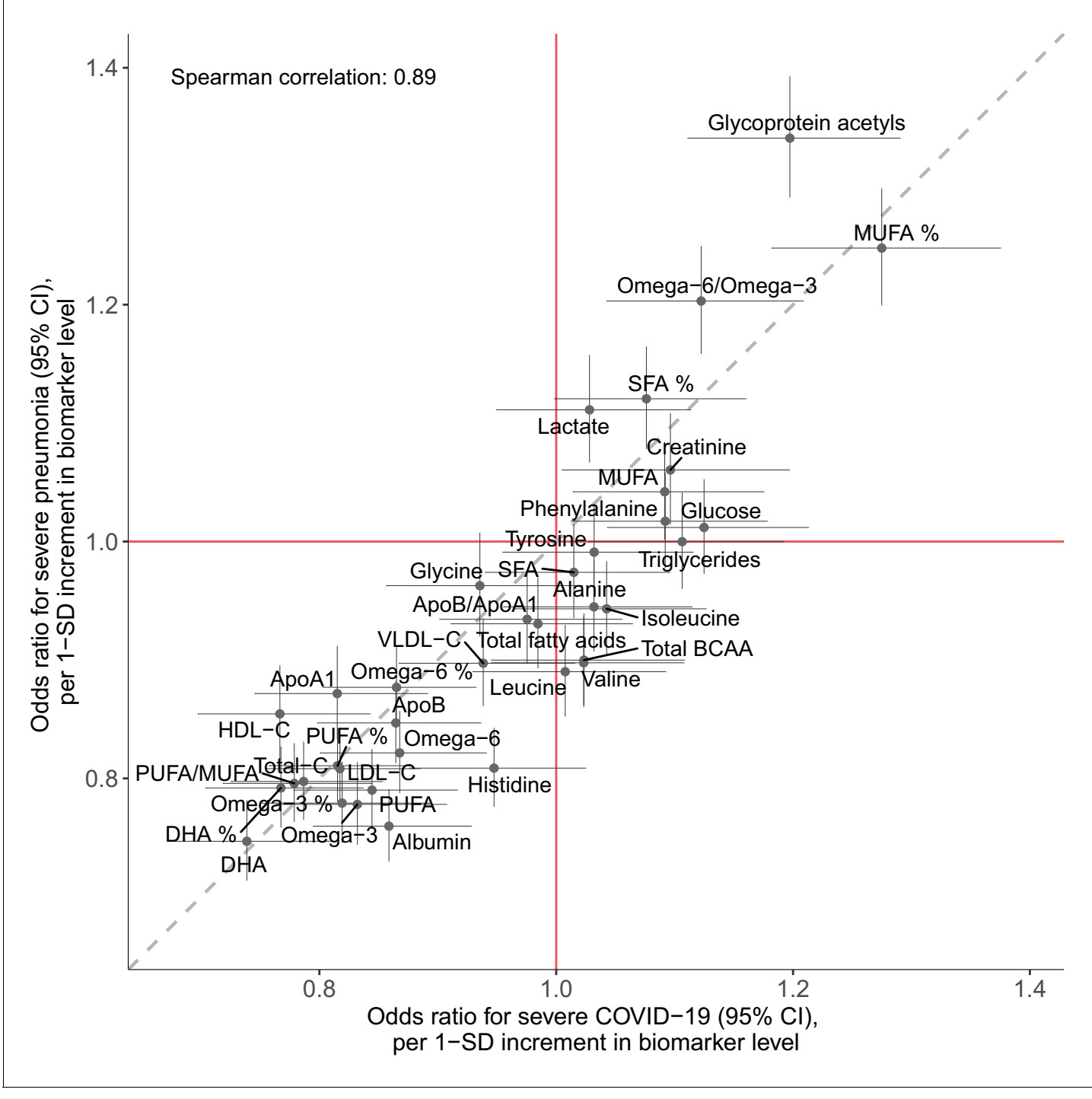

**Figure 7.** Concordance of the overall pattern of biomarker associations with future onset of severe pneumonia and severe COVID-19. Biomarker associations with future onset of severe pneumonia (y-axis) plotted against the corresponding associations with severe COVID-19 (x-axis). The odds ratios, with adjustment for age, sex, and assessment centre, for each of the 37 clinically validated biomarkers in the Nightingale Health NMR platform are given with 95% confidence intervals in vertical and horizontal error bars. The dashed line denotes the diagonal.

The online version of this article includes the following source data for figure 7:

**Source data 1.** Numerical tabulation of odds ratios, and 95% confidence intervals for results shown in *Figure 7*.

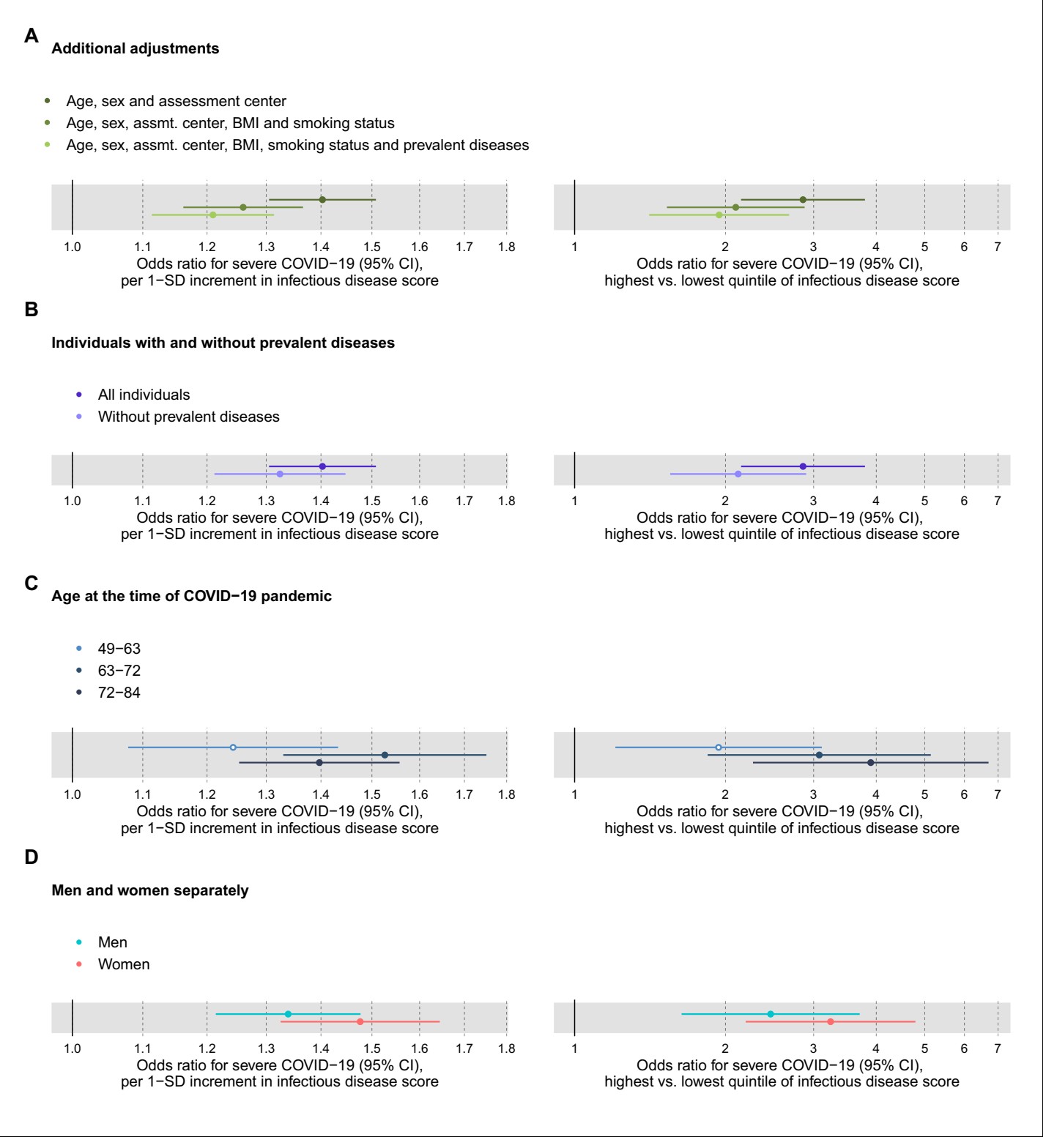

**Figure 8.** Relation of the multi-biomarker infectious disease score to future risk of severe COVID-19 with additional adjustments and in subgroups of the study population (n = 92,725; 652 cases diagnosed in hospital). (**A**) Odds ratios with severe COVID-19 after additional adjustments for BMI, smoking status and prevalent diseases. (**B**) Odds ratios with severe pneumonia in study participants with and without prevalent diseases at the time of blood sampling. (**C**) Odds ratios by age tertiles at the time of the COVID-19 pandemic. (**D**) Odds ratios for men and women, separately. The left-hand side shows the odds ratios per 1-SD increment in the multi-biomarker infectious disease score, and the right-hand side the odds ratios for comparing individuals in the highest and lowest quintiles of the score. All models are adjusted for age, sex, and assessment centre.

*Figure 8 continued on next page*

*Figure 8 continued*

The online version of this article includes the following source data for figure 8:

**Source data 1.** Numerical tabulation of odds ratios, betas, standard errors, and p-values for results shown in *Figure 8*.

propose that the overall metabolic biomarker perturbations observed here reflect molecular signals of low-grade inflammation that exacerbate disease severity, in case of both infectious and chronic diseases (*Akbar and Gilroy, 2020*; *Bonafè et al., 2020*). In line with this, prior studies have demonstrated that elevated levels of GlycA, the biomarker with the strongest weight in the infectious disease score, is associated with increased neutrophil activity and the long-term risk for fatal infections (*Ritchie et al., 2015*). Such over-activity of immune response from pneumonia or COVID-19 infection is known to cause tissue damage and organ dysfunction through cytokine storm, a common complication of severe COVID-19 (*Mangalmurti and Hunter, 2020*). While the specific biological mechanisms underpinning the blood metabolic biomarker associations with chronic and infectious diseases remain poorly understood, we emphasize that the observational character of our study does not allow us to conclude whether the biomarkers are contributing causally to increase the risk or are merely indirect risk markers.

Replication of novel biomarker associations is a key aspect in observational studies. We are not aware of other prospective studies with sufficient COVID-19 hospitalisation events and NMR-based metabolic biomarker data to address this. However, a preprint of the present study featured analysis of 195 severe COVID-19 cases, based on data available in UK Biobank back in June 2020 (*Julkunen et al., 2020*). In the present updated analyses, with over three times the number of cases, all biomarker associations with susceptibility to severe COVID-19 were similar or stronger, and

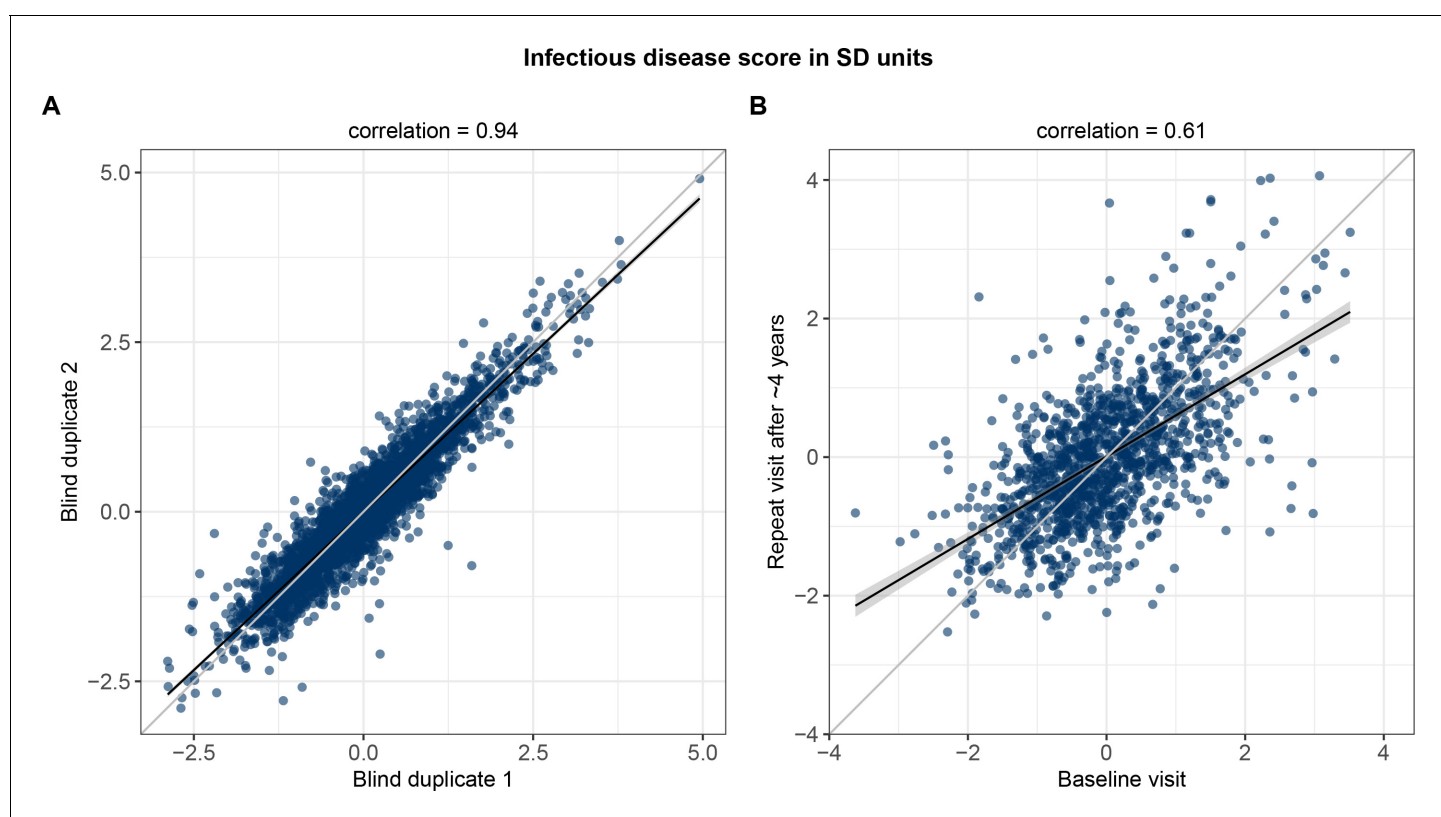

**Figure 9.** Technical repeatability for measuring the multi-biomarker infectious disease score and biological stability in repeat measures 4 years after the baseline blood sampling. (A) Technical repeatability of the infectious disease score assessed in blind duplicate samples. The correlation plot is based on 2863 blind duplicate plasma samples measured along with the regular measurements of ~105,000 samples in the Nightingale Health-UK Biobank initiative. (B) Biological stability of the infectious disease score based on plasma samples from 1298 individuals who attended both the baseline visit as well as a repeat visit ~4 years later at the UK Biobank assessment centres.

hereby provide a within-cohort replication of our initial findings. In addition, a recent study used the same metabolic biomarker panel in three cohorts of hospitalised patients and observed similar overall biomarker perturbations to be predictive of COVID-19 severity (*Dierckx et al., 2020*). The study also reported the multi-biomarker infectious disease score to be among the strongest biomarkers for discriminating COVID-19 severity among already hospitalised patients.

Our study has both strengths and limitations. Strengths include the large sample size, which enabled the analysis of biomarkers for susceptibility to severe COVID-19 based on pre-pandemic blood samples from general population settings. We used a validated metabolic profiling platform that enables simultaneous quantification of numerous metabolic biomarkers in a scalable low-cost setup. Although the number of hospitalised COVID-19 cases was in line with the prevalence in England, we acknowledge that the statistical power was limited for prediction analyses even with close to 100,000 samples linked with COVID-19 outcome data. Furthermore, the UK Biobank study participants are not fully representative of the UK population by demographic characteristics; the individuals were enrolled on a volunteer basis and are therefore more representative of healthier individuals than average (*Sudlow et al., 2015*; *Fry et al., 2017*). Even though this is generally not a concern for investigating risk associations (*Keyes and Westreich, 2019*), it does limit the statistical power to explore effects of ethnicity and old age. Other limitations include the decade long duration from blood sampling to the COVID-19 pandemic. While this limits inference on how well the biomarkers predict short-term risk for severe COVID-19, our analogy with long-term risk for severe pneumonia indicates that the time lag likely attenuates the biomarker association magnitudes substantially. Conversely, the remarkably strong associations for short-term risk of severe pneumonia led us to speculate that similar enhancements in association magnitudes could also hold for severe COVID-19. However, this inference should be further tested, in particular in the light of the bacterial origin of many severe pneumonia cases and the viral origin of COVID-19. Weaker biomarker associations for severe COVID-19 compared to severe pneumonia may also arise from the UK Biobank COVID-19 data being influenced by ascertainment bias in terms of differential healthcare seeking and differential testing (*Griffith et al., 2020*), whereas pneumonia is anticipated to have nearly complete case ascertainment (*Ho et al., 2020*).

In conclusion, a metabolic signature of perturbed blood biomarkers is associated with an increased susceptibility to both severe pneumonia and COVID-19 in blood samples collected a decade before the pandemic. The multi-biomarker score captures an elevated susceptibility to severe pneumonia within few years after blood sampling that is several times stronger than the risk elevation associated with many pre-existing health conditions, such as obesity and diabetes (*Ho et al., 2020*). If the three- to fourfold elevation in short-term risk compared to long-term risk of severe pneumonia also applies to severe COVID-19, then the metabolic biomarker profiling could potentially complement existing tools for identifying individuals most susceptible to a severe COVID-19 disease course. Regardless of the translational prospects, these results provide novel understanding on how metabolic biomarkers may reflect the susceptibility of severe COVID-19 and other infections.

## Materials and methods

### Study population

Details of the design of the UK Biobank have been reported previously (*Sudlow et al., 2015*). Briefly, the UK Biobank recruited 502,639 participants aged 37–70 years in 22 assessment centres across the UK. All study participants had to be able to attend the assessment centres by their own means, and there was no enrolment at nursing homes. All participants provided written informed consent and ethical approval was obtained from the North West Multi-Center Research Ethics Committee. Blood samples were drawn at baseline between 2007 and 2010. The current analysis was approved under UK Biobank Project 30418. No selection criteria were applied to the sampling.

### Metabolic biomarker profiling

From the entire UK Biobank population, a random subset of non-fasting baseline plasma samples (aliquot 3) from 118 466 individuals and 1298 repeat-visit samples were measured using high-throughput NMR spectroscopy (Nightingale Health Plc; biomarker quantification version 2020). This

provides simultaneous quantification of 249 metabolic biomarker measures in a single assay, including routine lipids, lipoprotein subclass profiling with lipid concentrations within 14 subclasses, fatty acid composition, and various low-molecular-weight metabolites such as amino acids, ketone bodies, and glycolysis metabolites quantified in molar concentration units. Technical details and epidemiological applications of the metabolic biomarker data have been reviewed (*Soininen et al., 2015*; *Würtz et al., 2017*). The Nightingale NMR platform has received various regulatory approvals, including CE-mark, and 37 biomarkers in the panel have been certified for diagnostics use. We focused on this particular set of certified biomarkers, as we wanted to investigate if these markers of systemic metabolism — commonly linked to cardiometabolic diseases — could also be associated with future risk for severe infectious disease. Furthermore, these clinically validated biomarkers span most of the different metabolic pathways measured by the NMR platform and could facilitate potential translational applications as they are certified for diagnostics use and are measured simultaneously in a single assay. The mean and standard deviation of concentrations for 249 quantified metabolic biomarkers are given in *Supplementary file 2*.

Measurements of the metabolic biomarkers were conducted blinded prior to the linkage to the UK Biobank health outcomes. The metabolic biomarker data were curated and linked to UK Biobank clinical data in late-May 2020. The metabolic biomarker dataset has been made available for the research community through the UK biobank in March 2021.

## Severe pneumonia outcomes

We combined ICD-10 codes J12–J18 to define the pneumonia endpoint. To strengthen the analogy with the analysis of severe COVID-19, we focused on severe pneumonia events, defined as diagnosis in hospital or death records based on UK Hospital Episode Statistics data and national death registries (2507 incident cases in the current study). All analyses are based on the first occurrence of a diagnosis. Therefore, 2658 individuals with recorded hospitalisation of pneumonia prior to the blood sampling were excluded. Additionally, 346 individuals with pneumonia diagnosis recorded in primary care settings and by self-reports were also omitted from the analyses. The registry-based follow-up was from blood sampling in 2007–2010 through to 2016–2017, depending on assessment centre (850,000 person-years).

## Severe COVID-19 outcomes

We used COVID-19 data available in the UK Biobank per 3rd of February 2021, which covers test results from 16 March to 1st of February 2021. These data include information on positive/negative PCR-based diagnosis results and explicit evidence in the microbiological record on whether the participant was an inpatient (*Resource UKBiobankD, 2020*). For the present analyses, we focused on PCR-positive inpatient diagnoses. These hospitalised cases are here denoted as severe COVID-19 (652 cases in the current study). COVID-19 data were not available for assessment centres in Scotland and Wales, so individuals from these centres were excluded. Individuals who had died during follow-up prior to 2018 were also excluded, since they were never exposed to COVID-19.

## Control group

The entire study population of non-cases was used as controls in the statistical analyses (n = 102,639 for severe pneumonia and n = 92,073 for severe COVID-19, respectively). This choice of controls is consistent with the majority of publications examining risk factors for susceptibility to severe COVID-19 (e.g. *Ho et al., 2020*; *Williamson et al., 2020*). It allows to address the question of whether an initially healthy person with a high value of a given biomarker is at an increased risk of eventually getting the disease outcome (severe pneumonia or COVID-19 hospitalisation) compared to people from the general population with low levels of the biomarker. This choice of controls also overcomes biases that may arise from analyses using confirmed mild infections as the control group, such as collider bias caused by non-random testing of the control group compared to the rest of the study population (*Griffith et al., 2020*).

## Prevalent diseases

To examine the influence of prevalent diseases in the prospective analyses of severe pneumonia and severe COVID-19, we used the following: prevalent cardiovascular disease (ICD-10 codes I20–I25,

I50, I60–I64, and G45), diabetes (E10–E14), lung cancer (C33–C34, D02.2, Z85.1), chronic obstructive pulmonary disease (COPD; J43–J44), liver diseases (K70–K77), renal failure (N17–N19), and dementia (F00-F03).

## Statistical methods

Biomarker levels outside four interquartile ranges from median were considered as outliers and excluded. All 37 biomarkers were scaled to standard deviation (SD) units prior to analyses. For biomarker association testing with severe pneumonia and with severe COVID-19 (as separate outcomes), we used logistic regression models adjusted for age, sex, and assessment centre. To examine the utility of multiple biomarkers in combination, we used a weighted sum of the biomarkers optimised for association with future risk of severe pneumonia; this multi-biomarker score was denoted as 'infectious disease score'. To minimise the collinearity of the biomarkers, the multi-biomarker score was trained using logistic regression with least absolute shrinkage and selection operator (LASSO), which uses L1 regularisation that adds penalty equal to the absolute value of the magnitude of the coefficients. The multi-biomarker infectious disease score was trained using half of the study population with complete data available for the 37 clinically validated biomarkers (n = 52,573 and 1257 severe pneumonia events) using five-fold cross-validation to optimise the regularizsation parameter λ. The remaining half of the study population was used in validating the performance of the biomarker score in relation to future risk for severe pneumonia. The multi-biomarker infectious disease score was subsequently tested for association with severe pneumonia and COVID-19 in logistic regression models adjusted for age, sex, and assessment centre. We further examined the effect of additional adjustment for body mass index (BMI) and smoking status (never, former, current) and prevalent diseases. The associations were also examined by omitting individuals with prevalent diseases and stratified by age and sex. In the case of severe pneumonia, we further examined the association magnitudes according to follow-up time: we used severe pneumonia events occurring during 7–11 years after the blood sampling to mimic the decade long lag from blood sampling to the COVID-19 pandemic, and severe pneumonia events occurring within the first 2 years to interpolate to the scenario of preventative COVID-19 screening carried out today. In both scenarios, the confined follow-up times were arbitrarily chosen to be as short as possible while ensuring sufficient numbers of events. Finally, to explore potential non-linear effects, the infectious disease score was plotted as a proportion of individuals who contracted severe pneumonia during follow-up when binning individuals into percentiles of the infectious disease score (*Khera et al., 2018*). The time-resolution was further examined by Kaplan-Meier curves of the cumulative risk for severe pneumonia.

## Acknowledgements

The authors are grateful to UK Biobank for access to data to undertake this study (Project #30418).

## Additional information

### Competing interests

Heli Julkunen: HJ is employee and holds stock options with Nightingale Health Plc. Anna Cichońska: AC is employee and holds stock options with Nightingale Health Plc. Peter Würtz: PW is employee and shareholder of Nightingale Health Plc. The other author declares that no competing interests exist.

### Funding

| Funder | Author |
| --- | --- |
| Nightingale Health Plc | Heli Julkunen<br>Anna Cichońska<br>Peter Würtz |

This work, including data collection, statistical analysis and writing of the paper, was done by employees of Nightingale Health Plc.

## Author contributions
Heli Julkunen, Conceptualization, Data curation, Software, Formal analysis, Investigation, Visualization, Methodology, Writing - original draft, Writing - review and editing; Anna Cichońska, Conceptualization, Data curation, Software, Formal analysis, Supervision, Investigation, Visualization, Methodology, Writing - original draft, Project administration; P Eline Slagboom, Investigation, Methodology, Writing - original draft, Writing - review and editing; Peter Würtz, Conceptualization, Supervision, Funding acquisition, Investigation, Visualization, Methodology, Writing - original draft, Project administration, Writing - review and editing

## Author ORCIDs
Heli Julkunen ⓘ https://orcid.org/0000-0002-4282-0248
Peter Würtz ⓘ https://orcid.org/0000-0002-5832-0221

## Ethics
Human subjects: The UK Biobank recruited 502 639 participants aged 37-70 years in 22 assessment centres across the UK. All participants provided written informed consent and ethical approval was obtained from the North West Multi-Center Research Ethics Committee. Details of the design of the UK Biobank have been reported previously (Sudlow et al PLOS Medicine 2015). The current analysis was approved under UK Biobank Project 30418.

## Decision letter and Author response
Decision letter https://doi.org/10.7554/eLife.63033.sa1
Author response https://doi.org/10.7554/eLife.63033.sa2

# Additional files

## Supplementary files
• Supplementary file 1. Table of weights of the biomarkers included in the multi-biomarker infectious disease score derived using LASSO regression. The table indicates the weights for the 25 biomarkers that were selected in derivation of the multi-biomarker infectious disease score, based on optimising prediction for severe pneumonia using logistic regression with LASSO in the derivation half of the study population (n = 52,573). Each biomarker was scaled to SD-units prior to the analyses. The infectious disease score was then calculated as $\beta_1 \times X_1 + \beta_2 \times X_2 + \ldots + \beta_{25} \times X_{25}$, with $X_i$ denoting the SD-standardised biomarker level for the $i$th biomarker and $\beta_i$ denoting the coefficient from the multi-biomarker logistic regression model. DHA indicates docosahexaenoic acid; MUFA: monounsaturated fatty acids; PUFA: polyunsaturated fatty acids; SFA: saturated fatty acids.

• Supplementary file 2. Mean biomarker concentrations and standard deviations, and odds ratios of all 249 biomarkers with severe pneumonia. The table indicates mean concentrations and standard deviations used for biomarker scaling. The table also includes numerical results of odds ratios of all 249 biomarkers with severe pneumonia, with corresponding 95% confidence intervals and p-values, and whether each biomarker is clinically validated and included in the multi-biomarker infectious disease score.

• Transparent reporting form

## Data availability
The data are available for approved researchers from UK Biobank. The metabolic biomarker data has been released to the UK Biobank resource in March 2021.

The following dataset was generated:

| Author(s) | Year | Dataset title | Dataset URL | Database and Identifier |
|---|---|---|---|---|
| Cichońska A, | 2021 | UK Biobank Nightingale biomarker | https://biobank.ndph.ox. | Biobank, 220 |

| Julkunen H, Würtz P | data | ac.uk/showcase/label.cgi?id=220 |

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
