## [Decision Letter]

**Acceptance summary:**

The authors show that in over 100,000 healthy individuals a baseline multi-biomarker score comprised of 25 proteins, fatty acids, amino acids and lipids obtained simultaneously from a single sample strongly predicts the risk of severe pneumonia and COVID 19 hospitalization 7-11 years later. These findings now been validated in Belgian patients provide novel insights for both pneumonia and severe COVID -19 risk.

**Decision letter after peer review:**

Thank you for submitting your article "Metabolic biomarker profiling for identification of high risk for severe pneumonia and COVID-19 hospitalisation" for consideration by *eLife*. Your article has been reviewed by 3 peer reviewers, including Edward D Janus as the Reviewing Editor and Reviewer #3, and the evaluation has been overseen by a Senior Editor. The following individual involved in review of your submission has agreed to reveal their identity: Harin Karunajeewa (Reviewer #2).

The reviewers have discussed the reviews with one another and the Reviewing Editor has drafted this decision to help you prepare a revised submission.

Summary:

The authors report that in over 100,000 healthy individuals a baseline multi-biomarker score comprised of 25 proteins, fatty acids, amino acids and lipids strongly predicts the risk of severe pneumonia and COVID 19 hospitalization 7-11 years later. The issue is whether other parameters related to inflammatory or immunological processes have been excluded. Also there are many simpler scores for determining severity and outcomes in clinical practice.

Thus there are two aspects the population susceptibility and the prediction of severe disease. There is a need to change the comparison group if looking for a useful predictor of severe disease.

The paper needs to be revised to highlight its hypothesis generation aspects which is its major point of interest, but also to cover or at least discuss the broader context of other inflammatory, immunological markers and other patient susceptibility characteristics and limitations including whether there is actual clinical utility.

Essential revisions:

This is a dense and complex analysis of a large, rich and powerful dataset that examines a clinically relevant issue that is obviously extremely timely. As an observational study it is limited to "hypothesis-generating conclusions". Its findings are intriguing and potentially of high medical and scientific importance.

The manuscript identifies a group of biomarkers that is highly predictive of severe pneumonia and COVID-19 hospitalization in prospective samples that were collected many years earlier. There are several strengths of the manuscript, including the large samples size using the UK biobank. The inquiry of a consistent set of markers that predict severe response to multiple important outcomes is also important. However, there are also not notable issues with the manuscript that need to be addressed and we believe inaccurate conclusions have been drawn from the analyses.

1. Intuitively one would expect markers of inflammation and immunological susceptibility to be the strongest predictors in this disease situation. Have these been explored previously? Is it that you are testing the already available marker set heavily slanted to metabolic markers because that is what is available so its wider use is being explored?

2. The primary concern with the manuscript as written is that the controls being used in both analyses do not seem appropriate. It is unclear why controls are being used in the analyses vs. those with pneumonia without a severe response or those with COVID-19 without a severe reaction. Aren't those the real comparison groups? By just using random controls, the analysis is not taking into account that disease itself. This is a fundamental flaw in the way the analyses are presented and the conclusions drawn. Furthermore, it completely inflates that actual statistical power in the analyses, given that the control group is literally several magnitudes bigger in the case of COVID-19.

3. Having said that, at an absolute minimum, these analyses should be rerun using COVID-19+ with severe and non-severe response. In this case, the real question is being answered: using prospective samples or individuals who get pneumonia/COVID, can we predict this that have a severe response vs. those who do not? In this case we are actually deciphering the difference between the biology of individuals who we know will have a severe response and those who we know will not. In the analyses, that the authors do, we have no idea who in the control group would have a severe response and who would not. Therefore the control group is completely inappropriate for addressing that issue.

4. In the case where one is using severe/non-severe, it should be acknowledged that the actual statistical power of the tru comparison is substantially lower that what is being used with the larger UK biobank samples.

5. Some approach to assess the robustness of these findings in an additional population would be ideal; however, it is acknowledged that this may not be feasible currently.

6. Please advise specifically on this point – that the metabolomic data has been transparently documented prior to the linked UK biobank data being unlocked, so that it can be matched up with that present in the final linked dataset – ie verifies that authors were blinded and hasn't changed since unblinded. The UK biobank policies and procedures hopefully address these issues of research integrity.

7. Please expand on clinical utility. The tests and score would need to be affordable and readily available and compliment existing predictors – not only cardiometabolic diseases but many others and specifically age and be cost effective.

A key line in the introduction outlines the study's rationale as its potential value for risk stratification as "facilitating the identification of high-risk individuals missed by current clinical tools". I think the authors should therefore address this point more explicitly when framing their results in the conclusion. To what extent do their results really support this statement? This could take the form of providing important context as to the predictive ability of existing or proposed clinical risk stratification tools. There has been an explosion of these in relation to COVID, many of which show that easily collected clinical and demographic information can predict very large differences in risk of hospitalization or death – eg Halalau et al. Annals of Internal Medicine quote an odds ratio of 19 for risk of hospitalization in their highest risk strata. So how much would metabolomic profiling really add to information much more readily and easily accessible without a blood test and expensive laboratory analysis?

8. The authors need to acknowledge the limitations of the clinical data that has been used to adjust comparisons. For instance they have adjusted for cardiac and respiratory disease but there is no mention of dementia, general "frailty, residence in a nursing home and measures of independent function which are all very highly associated with the risk of pneumonia. Could the high risk metabolic profiles lie on the same causal pathway and therefore just be surrogates for these easily-measured biological risk factors? Again, goes to the question of what metabolomic profiling could add to much more readily available clinical and demographic information.

---

## [Author Response]

Summary:The authors report that in over 100,000 healthy individuals a baseline multi-biomarker score comprised of 25 proteins, fatty acids, amino acids and lipids strongly predicts the risk of severe pneumonia and COVID 19 hospitalization 7-11 years later. The issue is whether other parameters related to inflammatory or immunological processes have been excluded. Also there are many simpler scores for determining severity and outcomes in clinical practice.Thus there are two aspects the population susceptibility and the prediction of severe disease. There is a need to change the comparison group if looking for a useful predictor of severe disease.The paper needs to be revised to highlight its hypothesis generation aspects which is its major point of interest, but also to cover or at least discuss the broader context of other inflammatory, immunological markers and other patient susceptibility characteristics and limitations including whether there is actual clinical utility.We thank the editors and reviewers for their valuable comments. We have now revised the paper according to the comments. We have also shifted the wording to emphasise the population susceptibility rather than the prediction aspect, and toned down some of the initially proposed public health implications to be more in line with current COVID-19 preventative policies and vaccine roll outs in many countries.We would like to emphasize two points that substantially strengthen the revised manuscript:1. We have re-analysed biomarker associations with severe COVID-19 based on updated COVID-19 data from UK Biobank. This update more than triples the number of severe COVID-19 cases (from 195 to 652). All association magnitudes are similar or stronger in these re-analyses, and the statistical significance is therefore substantially improved: all analyses of the multi-biomarker infectious disease score vs severe COVID-19 risk are now significant and many individual biomarkers are also now significant (Figure 6). The updated analysis thus provides a within-cohort replication of our initial results. The updated analysis also overcomes a prime limitation of the initial submission, since the multi-biomarker score associations with severe COVID-19 are now robust to adjustments and omission of individuals with prevalent diseases (Figure 8).2. After our initial submission, a medRxiv preprint (Dierckx et al. (2020), https://doi.org/10.1101/2020.11.09.20228221) has provided independent validation of our biomarker findings in hospital settings: this preprint examines the same metabolic biomarker panel in relation to COVID-19 severity in three cohorts of COVID‑19 patients in Belgian hospitals. Their results confirm that all the key changes in the metabolic biomarkers and the “infectious disease biomarker score” that we propose to be predictive of severe COVID-19 among initially disease-free individuals, are also discriminating COVID-19 severity among already hospitalised patients.Essential revisions:This is a dense and complex analysis of a large, rich and powerful dataset that examines a clinically relevant issue that is obviously extremely timely. As an observational study it is limited to "hypothesis-generating conclusions". Its findings are intriguing and potentially of high medical and scientific importance.The manuscript identifies a group of biomarkers that is highly predictive of severe pneumonia and COVID-19 hospitalization in prospective samples that were collected many years earlier. There are several strengths of the manuscript, including the large samples size using the UK biobank. The inquiry of a consistent set of markers that predict severe response to multiple important outcomes is also important. However, there are also not notable issues with the manuscript that need to be addressed and we believe inaccurate conclusions have been drawn from the analyses.1. Intuitively one would expect markers of inflammation and immunological susceptibility to be the strongest predictors in this disease situation. Have these been explored previously?

Many inflammatory biomarkers have been shown to predict COVID-19 severity among infected individuals, but only few studies have examined molecular markers of inflammation in relation to future COVID-19 risk among healthy individuals. This is likely due to the challenge of attaining the kind of study design we present here, as it requires that the biomarkers are measured from blood samples drawn prior to the pandemic. Studies with similar design based on UK Biobank have shown that hsCRP and blood cell counts are only modest predictors of COVID-19 hospitalisation and pneumonia (Ho et al., BMJ Open 2020); and those results are weaker than many of the metabolic biomarkers analysed in the present study (p. 7).

Glycoprotein acetyls (GlycA), the primary marker of low-grade inflammation in the metabolic biomarker panel analysed in our study, has indeed previously been associated with long-term risk for severe infections (Ritchie et al., Cell Systems 2015). Also PUFA% has been associated with risk of death from non-localised infections (Deelen et al., Nature Communications 2019), but no prior studies have examined the wider panel of metabolic biomarkers quantified by the NMR platform in relation to infectious diseases. Although GlycA was the strongest individual biomarker for the risk of severe pneumonia, this was not the case for the risk of severe COVID‑19 (for which blood levels of DHA and monounsaturated fatty acids were the strongest biomarkers). In addition to the expected results on the GlycA inflammatory biomarker, our study demonstrates that many cardiometabolic biomarkers, such as fatty acids and amino acids, can also be predictive of severe infectious disease risk among generally healthy individuals (p. 6). We believe that the novel evidence on how the susceptibility to severe infectious diseases is reflected in the systemic metabolic profile could broaden the understanding of impaired immunity beyond the focus on established inflammatory biomarkers. In addition, we show that the combination of many metabolic biomarkers measured simultaneously from a single assay provides stronger associations for infectious disease susceptibility than individual biomarkers.

Is it that you are testing the already available marker set heavily slanted to metabolic markers because that is what is available so its wider use is being explored?

All the metabolic markers analysed in our study are indeed obtained simultaneously in a single assay using NMR spectroscopy. We have now clarified this point in the manuscript (p. 3 and 7). The measurements of these metabolic biomarkers in >100,000 samples from the UK Biobank were undertaken prior to the COVID-19 pandemic. It is thus a pre-specified set of biomarkers that has been related primarily to risk for cardiovascular disease and diabetes in earlier epidemiological studies based on smaller cohorts.

The reason for focusing on this particular set of metabolic biomarkers was two-fold: first, we wanted to investigate whether these biomarkers of systemic metabolism, and commonly described as cardiometabolic risk markers, could be also associated with susceptibility to a severe infectious disease course in general population settings. The second reason was that these biomarkers can be measured simultaneously in a single blood assay at low cost. This allows using multi-biomarker scores for enhanced risk prediction without additional costs. These points can both provide molecular insights into COVID-19 etiology, and potentially open for novel tools to support identification of individuals who are most susceptible. The panel of biomarkers shown in Figures 2 and 6 are those from the employed NMR platform that have been certified for clinical use and which would therefore facilitate potential translational applications, as now noted on p. 3 and 7.

2. The primary concern with the manuscript as written is that the controls being used in both analyses do not seem appropriate. It is unclear why controls are being used in the analyses vs. those with pneumonia without a severe response or those with COVID-19 without a severe reaction. Aren't those the real comparison groups? By just using random controls, the analysis is not taking into account that disease itself. This is a fundamental flaw in the way the analyses are presented and the conclusions drawn. Furthermore, it completely inflates that actual statistical power in the analyses, given that the control group is literally several magnitudes bigger in the case of COVID-19.

The use of the entire non-case study population as controls in our statistical analyses is the established approach for biomarker association testing with disease risk in the general population settings – regardless of whether it is for infectious diseases, cardiovascular disease or any other disease. The same selection of controls is used by many other COVID-19 related publications based on UK Biobank (e.g. Petermann-Rocha et al., BMC Medicine 2020). The choice of using all non-cases as controls, rather than only test-positive mild cases, is also taken in high profile papers on risk factors associated with COVID-19-related death, e.g. the OpenSAFELY study on determining the relative risk ascribed to the prevalence of many chronic conditions (Williamson et al., Nature 2020).

We are convinced that the chosen control group and the statistical analysis framework is the appropriate one to address study question we are after: for a person with a given value of the biomarker score, what is the risk increase for the disease outcome (severe pneumonia or COVID-19 hospitalisation) compared to the risk in a reference population (such as people in the general population with median levels of that biomarker score). This is the same principle as when risk factors for COVID-19 mortality are reported, e.g. the two-fold risk increase linked to class III obesity *vs* people with normal weight (Williamson et al., Nature 2020). The chosen statistical analyses can thus be used to identify individuals who are at increased risk compared to their peers according to their metabolic biomarker profile. We have now revised the wording throughout the manuscript to emphasize that we estimate the susceptibility for the disease outcome with respect to that person’s peers, rather than the relative risk of a given person for getting severe disease vs mild disease. We have also clarified the selection of control group in detail on p. 8.

The statistical power in the analyses is dictated by the number of cases when there is strong class-imbalance between the number of cases and controls. We address this point further below in answer #4.

3. Having said that, at an absolute minimum, these analyses should be rerun using COVID-19+ with severe and non-severe response. In this case, the real question is being answered: using prospective samples or individuals who get pneumonia/COVID, can we predict this that have a severe response vs. those who do not? In this case we are actually deciphering the difference between the biology of individuals who we know will have a severe response and those who we know will not. In the analyses, that the authors do, we have no idea who in the control group would have a severe response and who would not. Therefore the control group is completely inappropriate for addressing that issue.

As reasoned above, we are answering a slightly different study question than whether an individual will have a severe response or not; instead, we deliberately address the study question of the elevated risk for severe disease relative to peers from the general population as the reference group. We have now framed this point as identification of individuals at increased susceptibility in the general population, e.g. in the title and the abstract of the revised manuscript.

The chosen study design was also deemed necessary due to the limited COVID-19 testing performed in the UK at the time of the analysis. At the time of submission, the COVID-19 testing was heavily focused on healthcare workers and those with severe symptoms. Therefore, the subset of UK biobank participants who were tested for COVID-19 were far from being a random sample of the study participants. This is known to cause bias and potentially spurious results, and it prevents us from robustly addressing the question of the risk for severe vs mild response for an infected individual. For instance, being a health care worker influences the likelihood of being tested at all, and in turn more healthcare workers than average end of being tested positive – if health care workers generally have more healthy biomarker profiles than mild cases who never get tested in the general population, it can inflate the differences between mild and severe cases.

The various biases introduced by different case-control designs of COVID-19 analyses in the UK Biobank have been described in detail recently (Griffith et al., Nature Communications 2020); we have now referred to this in the revised manuscript (p. 7 and 8). In the case of the suggested mild vs severe cases design the authors note: when sampling conditional on having a positive test for COVID-19 infection and testing is unlikely to be random, then conditioning on the positive test result introduces bias by factors causing severe infections as well as those causing increased likelihood of testing. The ascertainment bias induced by these factors would have therefore caused the associations to not reflect patterns in the general population of interest. Please also see our response above explaining the study question we aim to address with the chosen study design.

Finally, we note that the fraction of the population with severe COVID-19 in UK Biobank largely follow the hospitalisation rate in the UK, see Figure 1—figure supplement 1. Only a small fraction of hospitalised cases is likely to be missed (due to lack of inpatient positive PCR test); however, many mild cases continue to be missed throughout the pandemic, which further complicates addressing the study design and related scientific questions suggested by the reviewers.

4. In the case where one is using severe/non-severe, it should be acknowledged that the actual statistical power of the tru comparison is substantially lower that what is being used with the larger UK biobank samples.

Please see our response above explaining the reasoning behind the chosen study design.

We acknowledge that the statistical power is dictated by the number of cases rather than the total of the study population. We now further note this in the revised manuscript (p. 7).

The tripling of the number of severe COVID-19 cases in our analyses of the most recent UK Biobank alleviates this limitation. Nonetheless, we would like to emphasise that a sample size of >100,000 individuals is required to obtain sufficient number of cases in the prospective population-based study design that we present – and that this a major novelty compared to studies based on samples from already infected individuals.

5. Some approach to assess the robustness of these findings in an additional population would be ideal; however, it is acknowledged that this may not be feasible currently.

We agree that replication and validation of the findings are essential for biomarker studies. No replication data existed at the time of submission, owing to difficulties in accessing pre-pandemic blood samples with subsequent COVID-19 infection. This has now changed, with two sources providing replication and validation of our initial findings as detailed in a new paragraph on p. 6.

First, UK Biobank updates the information on COVID-19 hospitalisations on a monthly basis. This has more than tripled the number of cases with severe COVID-19 since the initial submission, from 195 to 652 individuals with metabolic biomarker data. When re-analysing the data using the enhanced number of severe COVID-19 cases, all association magnitudes are now similar or even stronger. This provides within-cohort replication of our results. Second, our findings have later on been supported by study on Belgian COVID-19 patient cohorts (Dierckx et al. 2020, MedrXiv preprint). The authors report strong associations with COVID-19 severity for the same set of metabolic biomarkers that we observe, e.g. inflammatory markers, amino acids and fatty acids. The multibiomarker score derived in the present study, “Infectious Disease Score”, was also found to be strongly associated with COVID-19 severity in these already infected patients (odds ratios of 4 and 4.5 with COVID-19 for having the most severe disease course, compared to mild-albeit-hospitalised disease course, in both patient cohorts with such data available). These results provide further support to the robustness of our findings.

6. Please advise specifically on this point – that the metabolomic data has been transparently documented prior to the linked UK biobank data being unlocked, so that it can be matched up with that present in the final linked dataset – ie verifies that authors were blinded and hasn't changed since unblinded. The UK biobank policies and procedures hopefully address these issues of research integrity.

The measurements of the metabolic biomarker data have been conducted blinded prior to linkage to the UK Biobank health outcomes. We have now clarified this point in the revised manuscript. We also emphasized that all the metabolic biomarker data analyzed for this paper are becoming available for the research community through UK Biobank, with data released on 23 March 2021 (p. 8 and 10).

We would like to note that the metabolic biomarker measurements have been undertaken in accordance with a prespecified protocol approved by UK Biobank’s biomarker advisory committee and UK Biobank laboratory. This includes detailed quality control assurance using blinded duplicate blood samples measured throughout the project. These aspects are detailed along with the release of the metabolic biomarker data for the research community by UK Biobank.

7. Please expand on clinical utility. The tests and score would need to be affordable and readily available and compliment existing predictors – not only cardiometabolic diseases but many others and specifically age and be cost effective.

We have toned down the wording on clinical utility throughout in the revised manuscript to account for the latest public health policy developments. Instead, we have emphasized the novel biological insights gained from the analyses of both pneumonia and severe COVID-19 risk.

The re-analyses of the COVID-19 data to the latest follow-up in UK Biobank, which more than tripled the number of cases with severe disease, reinforces the utility to identify high-risk individuals over and above existing risk factors. This includes age, which is accounted for in all the models. Nonetheless, the real-world clinical utility relies on the premise that the 3-fold increase in short-term risk observed for severe pneumonia, compared to long-term risk, also applies for risk of severe COVID-19. The inherently long follow-up time between UK Biobank blood sampling and COVID-19 prevents us from demonstrating this point with currently available data. We have acknowledged this limitation in the manuscript; further studies with shorter time between blood sampling and COVID-19 are under way, but accessing large amounts of samples with ascertained COVID-19 hospitalized and blood drawn prior to the pandemic is challenging.

With a view to translation, we would like to highlight that employed NMR platform has received various regulatory approvals, and the 37 biomarkers included in the present study have been certified for diagnostics use.

A key line in the introduction outlines the study's rationale as its potential value for risk stratification as "facilitating the identification of high-risk individuals MISSED BY CURRENT CLINICAL TOOLS". I think the authors should therefore address this point more explicitly when framing their results in the conclusion. To what extent do their results really support this statement? This could take the form of providing important context as to the predictive ability of existing or proposed clinical risk stratification tools. There has been an explosion of these in relation to COVID, many of which show that easily collected clinical and demographic information can predict very large differences in risk of hospitalization or death – eg Halalau et al. Annals of Internal Medicine quote an odds ratio of 19 for risk of hospitalization in their highest risk strata. So how much would metabolomic profiling really add to information much more readily and easily accessible without a blood test and expensive laboratory analysis?

We have now reworded the introduction and toned down the claims on potential clinical utility throughout, as described for the previous point. Despite the improved results in Figure 8 in the revised manuscript, the specific utility compared to other risk factors rely on the assumption of stronger associations for short-term risk of severe COVID-19, as demonstrated for short-term risk of severe pneumonia. We have also strongly clarified this point in the Abstract, Introduction and Discussion (p. 1, 2, 5 and 7).

We agree that many studies have suggested risk stratification tools for patients with acute COVID-19 infection, such as the cited paper by Halalau et al. That is very different from our study, which aims to identify high risk individuals *prior* to being infected, i.e. the susceptibility to a severe disease course in case they are infected. The risk differences are generally much smaller in this type of study setting. One of the most highly cited papers with such study setting is Williamson et al. Nature 2020, where they report hazard ratios of ~2-3 for many diseases that are commonly used to shift middle-aged people into the group of “high-risk individuals”, such as uncontrolled diabetes, class III obesity, and newly diagnosed cancer. The results we present on Figure 8 are of similar hazard ratios, and based on our hypothesis generating results for short-term risk of severe pneumonia, the odds ratios are likely to be considerably stronger magnitude for short-term risk.

8. The authors need to acknowledge the limitations of the clinical data that has been used to adjust comparisons. For instance they have adjusted for cardiac and respiratory disease but there is no mention of dementia, general "frailty, residence in a nursing home and measures of independent function which are all very highly associated with the risk of pneumonia. Could the high risk metabolic profiles lie on the same causal pathway and therefore just be surrogates for these easily-measured biological risk factors? Again, goes to the question of what metabolomic profiling could add to much more readily available clinical and demographic information.

We have now revised the analyses to also account for presence of dementia. The results were essentially unaltered because of this – in fact, none of the 652 severe COVID-19 cases had a dementia diagnosis at the time of blood sampling. This point reflects the general characteristics of the study population in the UK Biobank, with baseline age 40–70 and volunteer enrollment skewing the disease prevalence towards more healthy than average. All study participants had to be able to attend the assessment centres by their own means, and there was no enrollment at nursing homes. We have now clarified these points in the revised manuscript (p 4).

We agree that the metabolic biomarkers highlighted in our study correlate with ageing, and that they likely provide a molecular reflection of frailty, immune response and overall susceptibility to severe disease. However, the associations with future risk for pneumonia and severe COVID-19 were only weakly attenuated after accounting for prevalent cardiovascular disease, diabetes, lung cancer, COPD, liver diseases, renal failure and dementia – we are therefore convinced that they do reflect aspects of frailty and disease risk that is not easily captured by readily available clinical and demographic information.